# COVID-19 Vaccine Acceptability and Its Determinants in Mozambique: An Online Survey

**DOI:** 10.3390/vaccines9080828

**Published:** 2021-07-27

**Authors:** Janeth Dula, Alexandre Mulhanga, Armindo Nhanombe, Laurentino Cumbi, António Júnior, Joe Gwatsvaira, Joseph Nelson Siewe Fodjo, Edlaine Faria de Moura Villela, Sérgio Chicumbe, Robert Colebunders

**Affiliations:** 1Health Policy and Systems Program, National Institute of Health, Maputo 1120, Mozambique; janet.dula@ins.gov.mz (J.D.); alexandre.mulhanga@ins.gov.mz (A.M.); armindo.nhanombe@ins.gov.mz (A.N.); laurentino.cumbi@ins.gov.mz (L.C.); antonio.junior@ins.gov.mz (A.J.); sergio.chicumbe@ins.gov.mz (S.C.); 2Department of Mathematics and Statistics, University of Limerick, V94 T9PX Limerick, Ireland; joweygwatsv@gmail.com; 3Global Health Institute, University of Antwerp, 2000 Antwerp, Belgium; JosephNelson.SieweFodjo@uantwerpen.be; 4Disease Control Coordination, São Paulo State Health Department, São Paulo 01246-000, Brazil; edlaine@alumni.usp.br; 5Institute of Tropical Pathology and Public Health, Federal University of Goiás, Goiânia 74605-050, Brazil

**Keywords:** SARS-CoV-2, COVID-19, vaccine acceptability, Mozambique, online survey

## Abstract

A high worldwide SARS-CoV-2 vaccine coverage must be attained to stop the COVID-19 pandemic. In this study, we assessed the level of willingness of Mozambicans to be vaccinated against COVID-19. Data were collected between 11 and 20 March 2021, through a self-administered online survey. Of the 1878 respondents, 30.1% were healthcare workers, 58.3% were aged between 18 and 35 years, 60% were male, and 38.5% were single. Up to 43% had been tested for COVID-19 and 29% had tested positive. Overall vaccine acceptability was 71.4% (86.6% among healthcare workers, 64.8% among other respondents; *p* < 0.001). Reasons for vaccine hesitancy included: fear of vaccine side effects (29.6%) and the belief that the vaccine is not effective (52%). The acceptability of the SARS-CoV-2 vaccine increased with increasing vaccine efficacy. Using logistic regression, determinants for acceptability of the vaccine were: older age, a past COVID-19 test, a concern of becoming (re)infected by COVID-19, having a chronic disease, and considering vaccination important for personal and community health. In conclusion, vaccine acceptability in Mozambique was relatively high among healthcare workers but significantly lower in the rest of the population. This suggests that there is a need to educate the general population about SARS-CoV-2 vaccination and its importance.

## 1. Introduction

The new coronavirus (SARS-CoV-2) has caused a pandemic that destabilized health, economy, and circulation of individuals worldwide. The first COVID-19 cases were detected in China and the pandemic was declared in March 2020 [1]. By 12 June 2021, more than 176 million people had been infected worldwide [2]. The African continent remains the least affected, with a little over 5.0 million confirmed cases and 134,809 deaths. In Mozambique, 71,461 confirmed cases of COVID-19 and 837 deaths have been reported [2].

Current measures against the disease aim to control transmission and include mainly individual and community actions of improved hand hygiene, physical distancing, and the use of face masks. However, a large-scale COVID-19 vaccination campaign across the globe seems to be the only way out of this epidemic. Benefits of vaccination are multiple: disease prevention, reduction of the severity and mortality of the disease, reduction of the impact of the pandemic on the National Health System and the economy, mainly by protecting the most vulnerable groups [3]. As of 12 June, more than 2.36 billion vaccine doses have already been administered worldwide [4]. However, there is a stark gap between vaccination programs in different countries. As of 12 June, in Mozambique only 0.78% of the population received a first dose of a COVID-19 vaccine [2]. The process of supplying vaccines to low- and middle-income countries remains a challenge. Difficulties in increasing production and promoting equitable distribution of the SARS-CoV-2 vaccines continue to be the main barrier to ending the pandemic [5].

The World Health organization adopted the Principles and Values Framework of the Strategic Advisory Group of Experts on Immunization (SAGE) as the basis for the worldwide roll-out of the COVID-19 vaccination campaign. These principles are based on equal respect, global equity, national equity, reciprocity, and legitimacy [5,6]. The SAGE proposes a roadmap for prioritizing uses of COVID-19 vaccines that considers priority populations for vaccination based on epidemiologic setting and vaccine supply scenarios. This roadmap builds on the WHO SAGE values framework for the allocation and prioritization of COVID-19 vaccination. The rationale is minimization of the impact of the pandemic on the main risk groups, the healthcare system, and society at large. Healthcare workers, people in high-transmission environments, and people with medical conditions such as diabetes, lung disease, heart disease, and obesity have been associated with worse COVID-19 disease outcomes [5,6] and therefore should be vaccinated first.

In Mozambique, little is currently known about the acceptability of the SARS-CoV-2 vaccine by the population. The population’s perception of the vaccine will most likely be influenced by their knowledge about the health consequences of SARS-CoV-2 infection and the importance of the vaccine to prevent these consequences. This will in turn determine vaccine acceptability by the population. Conducting this survey to investigate the perceptions of Mozambicans about COVID-19 vaccines and their willingness to be vaccinated is important to plan the COVID-19 vaccination campaign in Mozambique.

## 2. Material and Methods

### 2.1. Study Setting and Design

A cross-sectional, self-administered electronic survey was conducted from 11 to 20 March 2021, among the general population of Mozambique. The study was part of a series of online surveys organized by the International Citizen Project COVID-19 (ICPcovid; online platform available at: https://www.icpcovid.com/en/covid-19-vaccination (accessed on 22 July 2021) to investigate the acceptability of vaccines against COVID-19. The English version of the ICPcovid online questionnaire was translated into Portuguese and adapted to the local context by the Mozambican research team (Appendix A). The survey was launched by a team of health professionals from the National Institute of Health in Mozambique. A snowball sampling approach was used, by sharing the questionnaire through WhatsApp, institutional web pages (including the one for the Ministry of Health of Mozambique), and other social networks, while asking participants to distribute the questionnaire to their contacts.

### 2.2. Data Collection

Individuals aged 18 years and above who gave their informed e-consent (by ticking an electronic checkbox) were able to participate in the survey. We collected socio-demographic data, information about underlying health conditions, acceptability of the COVID-19 vaccine, and associated factors such as vaccine efficacy, vaccine origin, and brand of the vaccine.

A 5-point Likert scale was used to assess the participant’s concern about becoming (re)infected with SARS-CoV-2 during the ongoing outbreak and determinants for acceptability; scores ranged from 1 (minimum level of fear, concern, or difficulty) to 5 (highest level). All responses were sent anonymously to the ICPcovid platform, stored on a password protected server in Belgium until the data were extracted for analysis.

### 2.3. Data Analysis

Descriptive statistics were presented as percentages (%) for categorical variables; continuous variables such as age were also categorized into age groups. A multiple logistic regression model was built to investigate factors associated with acceptability of the vaccine. The dependent variable was “vaccine acceptability: Yes versus no” [7]. Age and gender were introduced into the model to account for the population’s demographics, and all other covariates which had a *p*-value below 0.25 during univariate analysis were also included in the final model.

### 2.4. Ethical Considerations

The study protocol was approved by the Institutional Bioethics Committee of the National Institute of Health of Mozambique (Ref: 31/CIBS-INS/2021) and by the Ethics Committee of the University of Antwerp (Ref: 20/13/148).

## 3. Results

### 3.1. Socio-Demographic Characteristics of the Participants

The responses of 1878 participants were included in the analysis. Most respondents were in the 18–35 age group (58.3%), the majority were male (60%), and 38.5% were single (Table 1).

### 3.2. Participants’ Health Characteristics

Overall, 43% (808) had been tested for COVID-19, and in 28.8% (233), the test result was positive. More healthcare workers had been COVID-19 tested (57.2%) compared to non-healthcare workers (37.0%); *p* < 0.001. Among respondents who reported COVID-19 test results, positivity rate was similar among the 324 healthcare workers (28.4%) and the 485 non-healthcare workers (29.1%); *p* = 0.854. Six hundred twenty-three (33.2%) participants were reported to suffer from a chronic disease (Table 2).

### 3.3. Acceptability of the Vaccine against COVID-19

Of all the study participants, 1340 (71.4%) would agree to take the SARS-CoV-2 vaccine. Acceptance was higher among healthcare workers (86.6%) compared to the general population (64.9%); *p* < 0.001 (Table 3). Reasons for vaccine hesitancy included: belief that the vaccine is not effective (52.0%); fear of side effects of the vaccine (29.6%); beliefs that the vaccine was made to cause harm (1.8%). To consider accepting vaccination against SARS-CoV-2, 204 (15.2%) would consider the brand (manufacturer), 320 (23.9%) its country of origin, 559 (41.7%) other aspects, and 258 (19.3%) did not have an opinion on what to consider before accepting the vaccine. We also observed that the greater the reported efficacy of the vaccine, the more respondents would be likely to accept it. Of the participants who would accept the vaccine, 38.7% would accept the vaccine if it were 50% to 60% efficacious, while up to 89.5% would accept the vaccine if it were at least 90% efficacious (Table 3).

### 3.4. Factors Associated with Acceptability of the Vaccine against COVID-19

In multivariable analysis, subjects above 60 years and healthcare workers were more likely to accept the SARS-CoV-2 vaccine. Irrespective of the COVID test result, participants who had previously been tested for COVID-19 were more willing to take the vaccine compared to those who were never tested (Table 4). Moreover, participants who were “extremely concerned” about COVID-19 (re)infection and those who suffered from a chronic disease had higher odds of accepting the SARS-CoV-2 vaccine. Finally, participants who considered vaccination important for their own health or for the health of the community were more willing to accept vaccination, compared to those who considered vaccination not or less important.

## 4. Discussion

High worldwide SARS-CoV-2 vaccine coverage is required to attain a sufficiently high population level of “herd immunity” that would stop the COVID-19 pandemic. This means that in the African continent (including Mozambique), high SARS-CoV-2 vaccine coverage would also need to be reached. Our online survey showed an overall vaccine acceptability rate of 71.4% among Mozambican study participants. Acceptability was relatively high among healthcare workers (86.6%) but significantly lower in the general population (64.8%). Such higher vaccine acceptance among healthcare workers was also reported in a recent survey in Somalia [8] and the Eastern Cape in South Africa [9], but surprisingly in the Democratic Republic of Congo (DRC), healthcare workers were less willing to receive a COVID-19 vaccine [10]. In a recent systematic review of COVID-19 vaccine hesitancy worldwide, in 62% of the surveys among the general public, the acceptance of COVID-19 vaccination was at least 70% [11]. However, low rates of COVID-19 vaccine acceptance were reported in the Middle East (for example in Kuwait (23.6%), Jordan (28.4%), and Saudi Arabia (64.7%), Russia (54.9%), and certain African and European countries [10,12,13]. In Qatar, a Nationally Representative Survey of Qatari Nationals and Migrants conducted between December 2020 and January 2021 using phone interviews showed that 42.7% (95% CI: 39.5–46.1) accepted being vaccinated with a COVID-19 vaccine, 45.2% (95% CI: 41.9–48.4) were hesitant, and 12.1% (95% CI: 10.1–14.4) were resistant [14]. In South Africa, vaccine acceptance rates were assessed in nine surveys showing rates varying between 52 and 82% [15]. In other African countries, vaccine acceptance rates were 88% in Egypt [16] and 74.5% in Nigeria [17] but only 54.1% in Ghana [18], 53.6% in Uganda [19], and 46.1% in Ethiopia [20]. Of the eight studies conducted among healthcare workers, three surveys reported vaccine acceptance rates below 60%, with the lowest acceptance rate in the DRC (27%) [11,21]. The higher vaccine acceptance rate among healthcare workers in Mozambique is a positive finding, as it has been shown that healthcare workers could increase the confidence of the general population in the vaccine by informing them about its safety and efficacy. When healthcare professionals are vaccine-confident and vaccinate themselves, they are more likely to recommend vaccination to their patients.

Our study found that vaccine acceptability increased with vaccine efficacy, corroborating recent findings from an international survey [22]; the latter was conducted in several other low- and middle-income countries, and revealed that for a COVID-19 vaccine that is 90% effective, the vaccine acceptance rate in Brazil was 88.9%, in Thailand 58.5%, in Malaysia 55.4%, in Uganda 65.4%, but was very low in many other African countries, such as 32.9% in the DRC, and 22.6% in Benin [22]. Thus COVID-19 vaccine hesitancy is a global phenomenon complicating the vaccine roll-out in different countries and continents.

Our study showed that the lower the age, the lower the vaccine acceptability. This was also found in Saudi Arabia [23] and South Africa [15]. This is an important finding, as advanced age is a risk factor for severe COVID-19 disease and, therefore, vaccine acceptability by the elderly is crucial to prevent severe disease, hospitalizations, and death [5,24]. However, in a study in Ghana, persons 36–45 years of age were less likely to accept being vaccinated compared to those aged 18–25 years [18]. Once again, this shows that acceptability varies by country, region, culture, and that some determinants differ depending on context. In the same light, suffering from a chronic disease was a determinant for increased vaccine acceptability as was observed in Egypt but not in several other low- and middle-income countries [22].

Participants who considered vaccination important for their own health or for the health of the community were more willing to accept vaccination. This suggests that if more people are educated about the importance of vaccination against COVID-19, their willingness to be vaccinated will increase. Our findings are similar to an online study conducted in China which showed that 76.6% of the participants believed the vaccine would be beneficial for their health [25]. Gender, marital status, and the result of the COVID-19 test were not associated with vaccine availability as was reported in a survey in the DRC [10]. However, a recent survey in several low- and middle-income countries [22] and a survey in Somalia [8] showed that being a female reduced the odds for COVID-19 vaccine acceptance [8,22]. Moreover, in a systematic review of nationally representative samples in high-income countries, being male, older, high-income, highly educated, and belonging to a majority ethnic group were consistently associated with greater likelihood of having the intention to be vaccinated [26].

Increasing COVID-19 vaccine acceptance will require involving many stakeholders. Non-governmental and civil society organizations as well as influential faith and cultural leaders, could play a crucial role in motivating communities to increase acceptance of COVID-19 vaccination. To reach high COVID-19 vaccination coverage in Africa will be challenging. However, this COVID-19 pandemic is also a unique opportunity to demonstrate the importance of vaccination for public health.

## 5. Study Limitations

Our study population cannot be representative of the general population of Mozambique. Indeed, individuals without access to the internet (i.e., people with less education and belonging to lower social classes) did not have the opportunity to participate in the survey. Many studies have documented that a lower level of education was associated with lower vaccine acceptance [9,13,17,27,28,29]. Moreover, a large proportion of the respondents were healthcare workers. The latter is most likely the effect of the way the survey was launched, mainly by investigators with a medical background and using a snowballing approach. We therefore presented stratified results to disentangle the responses from healthcare workers and other respondents. Moreover, not everybody in Mozambique was aware of the survey. Similar to the survey conducted in high-income countries [26], it is important that, in low-income countries as well, vaccine acceptability studies should be organized on a representative sample of the population. We must also emphasize the cross-sectional design of our study, which implies that we only present a snapshot of the situation at the time of the survey. Perceptions about vaccines can change rapidly and are influenced by social media among other factors. It is important to note that our survey was carried out before the rare side effects of the AstraZeneca and the J&J vaccine became known and were widely publicized.

## 6. Conclusions

This survey in Mozambique showed that vaccine acceptability was relatively high among healthcare workers but significantly lower in the rest of the population. SARS-CoV-2 vaccination is still to start in Mozambique. COVID-19 vaccine roll-out programs must engage all stakeholders to ensure that communities are correctly informed. In Mozambique it is recommended to intensify the mobilization for COVID-19 vaccination, conveying positive messages regarding the efficacy and side effects of the vaccine to reduce fear and increase acceptability. It is expected that the rate of vaccination hesitancy will decrease once the population experiences the vaccination rolling out as being well organized and that people do not experience serious side effects. Currently, as in many other African countries, Mozambique is facing a serious third COVID-19 wave driven by the Delta variant [30]. However, less than 7000 COVID-19 vaccines are administered per day [31]. This rate will have to be increased quickly to reduce COVID-19 related mortality.

## Figures and Tables

**Table 1 vaccines-09-00828-t001:** Socio-demographic characteristics of the participants.

Characteristics	Number (%) *N* = 1878
Age	
18–35 years	1095 (58.3%)
36–60 years	719 (38.3%)
>60 years	64 (3.4%)
Sex	
Female	750 (40.0%)
Male	1128 (60.0%)
Marital Status	
Single	723 (38.5%)
Married	680 (36.2%)
Cohabitation	403 (21.6%)
Other	72 (3.8%)

**Table 2 vaccines-09-00828-t002:** Participants’ health characteristics.

Characteristics	Number (%) *N* = 1878
Tested for COVID-19 (entire sample)	
Yes	809 (43.0%)
No	1069 (57.0%)
Tested for COVID-19 (healthcare workers)	
Yes	324 (57.2%)
No	242 (42.8%)
Tested for COVID-19 (other participants)	
Yes	485 (37.0%)
No	827 (63.0%)
Test result for COVID-19 (entire sample)	
Positive	233 (28.8%)
Negative	537 (66.4%)
Unknown	39 (4.8%)
Test result for COVID-19 (healthcare workers)	
Positive	92 (28.4%)
Negative	216 (66.7%)
Unknown	16 (4.9%)
Test result for COVID-19 (other participants)	
Positive	141 (29.1%)
Negative	321 (66.2%)
Unknown	23 (4.7%)
Chronic health conditions	
Heart disease	32 (1.7%)
Hypertension	207 (11.0%)
Cancer	5 (0.3%)
HIV	54 (2.9%)
Tuberculosis	3 (0.2%)
Respiratory diseases	147 (7.8%)
Stroke	2 (0.1%)
Other	173 (9.2%)
I do not suffer from any chronic disease	1257 (66.9%)

**Table 3 vaccines-09-00828-t003:** Acceptability of the vaccine against COVID-19.

Characteristics	Number (%) Participants *N* = 1878
Vaccine acceptability (entire sample)	
Yes	1340 (71.4%)
No	538 (28.6%)
Vaccine acceptability (healthcare workers)	
Yes	490 (86.6%)
No	76 (13.4%)
Vaccine Acceptability (Other participants)	
Yes	850 (64.8%)
No	462 (35.2%)
Reasons for vaccine hesitancy +	
I’m afraid of the side effects of the vaccine	159 (29.6%)
I think the vaccine is not effective	280 (52.0%)
I think the vaccine was made to harm us	34 (6.3%)
My body is naturally strong, I don’t need a vaccine	6 (1.1%)
Respiratory diseases	2 (0.4%)
I already had COVID-19, so I think I am immune to the disease	1 (0.2%)
I do not think COVID-19 exists	7 (1.3%)
Other reason	49 (9.1%)
What would determine your vaccine acceptance *	
Vaccine origin	320 (23.9%)
Vaccine brand	204 (15.2%)
Other aspects	559 (41.7%)
No opinion	258 (19.3%)
Vaccine acceptability according to vaccine efficacy *	
50% to 60%	518 (38.7%)
60% to 70%	648 (48.4%)
70% to 80%	879 (65.6%)
80% to 90%	1050 (78.4%)
90% +	1199 (89.5%)

+ Percentages are calculated with the total number of participants who would not accept the vaccine as denominator. * Percentages are calculated with the total number of participants who would accept the vaccine as denominator.

**Table 4 vaccines-09-00828-t004:** Multiple logistic regression model investigating predictors for COVID-19 vaccine acceptability.

Covariate	Adjusted OR (95% CI)	*p*-Value
Age		
>60 years	Ref	
36–60 years	0.30 (0.09–0.83)	0.035
18–35 years	0.22 (0.06–0.61)	0.008
Sex		
Male	Ref	
Female	0.80 (0.60–1.06)	0.122
Marital status		
Single	Ref	
Married	1.31 (0.92–1.86)	0.136
Other	0.80 (0.38–1.78)	0.572
Cohabitation	0.72 (0.50–1.05)	0.083
Type of respondents		
Healthcare workers	Ref	
Non-healthcare workers (general population)	0.31 (0.22–0.44)	<0.001
Previously tested for COVID-19		
Yes	Ref	
No	0.63 (0.45–0.88)	0.008
What was the test result?		
Negative	Ref	
Positive	0.75 (0.46–1.23)	0.248
Unknown	1.14 (0.41–3.79)	0.811
Level of worry /concern about becoming (re)infected with the SARS-CoV-2 virus (Likert scale)		
Extremely concerned	Ref	
Very concerned	1.04 (0.79–1.37)	0.784
Moderately concerned	0.63 (0.44–0.91)	0.014
A little concerned	0.71 (0.43–1.16)	0.162
Not at all concerned	0.29 (0.19–0.45)	<0.001
Chronic disease		
I do not suffer from chronic disease	Ref	
I suffer from chronic illness	1.25 (1–1.55)	0.048
Importance of vaccination for your health		
Important	Ref	
Moderate	0.62 (0.50–0.92)	0.012
Not important	0.04 (0.03–0.05)	<0.001
Importance of vaccination for community health		
Important	Ref	
Moderate	0.71 (0.53–0.95)	0.021
Not important	0.05 (0.04–0.06)	<0.001

OR: odds ratio; CI: confidence interval; Ref: reference category.

## Data Availability

The data presented in this paper are available upon reasonable request to the corresponding author.

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
