# Peer review of "COVID-19 Vaccine Acceptability and Its Determinants in Mozambique: An Online Survey"

_vaccines, 2021, doi:10.3390/vaccines9080828_

Round 1

Reviewer 1 Report

This paper examines, via on line surveys, the acceptance of COVID-19 vaccination in Mozambique by respondents. 

The respondent population is a bit heavy on healthcare workers.  But, this may be a limitation due to demographics in this country. 

Generally, I find the paper to be OK.  It is of interest that vaccine efficacy vs. possible side effects seem to be major factors in acceptance considering the large amount of press releases on these topics in western countries. 

This would seem to indicate some rationale for acceptance that might be countered with better information. 

I do recommend altering the reference to "lower social classes" on line 198 (page 6, bottom), to terms used in other parts of the paper, like "less educated" and/or "lower income groups".

Author Response

Response Reviewer 1

This paper examines, via on line surveys, the acceptance of COVID-19 vaccination in Mozambique by respondents. 

The respondent population is a bit heavy on healthcare workers.  But, this may be a limitation due to demographics in this country. 

Response

The high number of health care workers is a consequence of the way the survey was launched, initially by a team of health professionals from the national institute of health in Mozambique.

Generally, I find the paper to be OK.  It is of interest that vaccine efficacy vs. possible side effects seem to be major factors in acceptance considering the large amount of press releases on these topics in western countries. 

This would seem to indicate some rationale for acceptance that might be countered with better information. 

I do recommend altering the reference to "lower social classes" on line 198 (page 6, bottom), to terms used in other parts of the paper, like "less educated" and/or "lower income groups"

Response

We dropped the term “lower social classes”

Reviewer 2 Report

Dula et al performed and analyzed an online survey to obtain information on the acceptibility of COVID-19 vaccine in Mozambique.

My major concern is that I do not think that the data shown in this study are representative for the entire population of Mozambique. On the one hand, 1879 participants represent only a small proportion of the entire population of Mozambique. On the other hand, as the authors stated, the participation relied on exclusion criteria such as missing access to the internet, inability to read etc. Further, most likely not every citizen has been informed about this online survey. Therefore, the participants are already some kind of preselected and might not be representative for Mozambique. This should be stated more clearly and not only in the last paragraph "study limitations".

Minor points:

  1. The presentation of the data shown in Tab. 2 might be misleading. In the text is it mentioned that 33.1% of the participants reported to suffer from chronic diseases, in particular hypertension (33.3% of all chronic diseases) or respiratory disease (23.6% of all chronic diseases). The numbers given in the table are set in relation to the total number of participants, therefore the percentages are 1/3 of the numbers given in the text. There is no information in the text or the table which numbers refer to the total number of participants and which of them to a selected group within the total number.
  2. As I could not find any appendix showing the survey questionnaire, it is unclear whether the participants were able to pick more than one answer regarding reasons for vaccine gesitancy or whether they had to choose the most likely answer.

Author Response

Reviewer 2

Dula et al performed and analyzed an online survey to obtain information on the acceptibility of COVID-19 vaccine in Mozambique.

My major concern is that I do not think that the data shown in this study are representative for the entire population of Mozambique. On the one hand, 1879 participants represent only a small proportion of the entire population of Mozambique. On the other hand, as the authors stated, the participation relied on exclusion criteria such as missing access to the internet, inability to read etc. Further, most likely not every citizen has been informed about this online survey. Therefore, the participants are already some kind of preselected and might not be representative for Mozambique. This should be stated more clearly and not only in the last paragraph "study limitations".

Response

We agree our study participants were not representative of the general population in Mozambique. We now added in the methods that the survey was launched by a team of health professionals from the national institute of health in Mozambique.

We now stressed this in the limitations. Moreover not everybody in Mozambique was aware of the survey.

Minor points:

  1. The presentation of the data shown in Tab. 2 might be misleading. In the text is it mentioned that 33.1% of the participants reported to suffer from chronic diseases, in particular hypertension (33.3% of all chronic diseases) or respiratory disease (23.6% of all chronic diseases). The numbers given in the table are set in relation to the total number of participants, therefore the percentages are 1/3 of the numbers given in the text. There is no information in the text or the table which numbers refer to the total number of participants and which of them to a selected group within the total number.

Response

Thanks for the question. We corrected the values in the text, as the correct values are shown in the table2.

  1. As I could not find any appendix showing the survey questionnaire, it is unclear whether the participants were able to pick more than one answer regarding reasons for vaccine hesitancy or whether they had to choose the most likely answer.

Response

The participants were able to pick the answers from a list in the questionnaire. We now include the questionnaire as a supplement

Reviewer 3 Report

The manuscript describes a cross-sectional survey on the acceptance of COVID vaccination in Mozambique. The survey was conducted through an online questionnaire and distributed through a "snowball" system. Most of the respondents were health workers. The overall willingness to be vaccinated was high at about 65%, but even higher among health workers.
I can only point out a few points in the presentation of the results that should be improved. The rest of the manuscript is successful. This is especially true for the honest description of the limitations of the study. 

Minor points:

Please indicate if you did any checks on the data before conducting the multiple logistic regression model and if not, why you think that these usual checks were not necessary. If you did, please include the brief description in the manuscript and state "results not presented".

Please check the order of presentation in table 2. The specified groups (healthcare workers) should appear above (before) the unspecified groups (others).

In table 3, the percentages in "vaccine acceptability according to vaccine efficacy" does not refer to the total of the table but to the total of answering "yes" to acceptance of vaccine. Please make this clearer to the reader.    

Author Response

Reviewer 3

The manuscript describes a cross-sectional survey on the acceptance of COVID vaccination in Mozambique. The survey was conducted through an online questionnaire and distributed through a "snowball" system. Most of the respondents were health workers. The overall willingness to be vaccinated was high at about 65%, but even higher among health workers.
I can only point out a few points in the presentation of the results that should be improved. The rest of the manuscript is successful. This is especially true for the honest description of the limitations of the study. 

Minor points:

Please indicate if you did any checks on the data before conducting the multiple logistic regression model and if not, why you think that these usual checks were not necessary. If you did, please include the brief description in the manuscript and state "results not presented".

Response

Please check the order of presentation in table 2. The specified groups (healthcare workers) should appear above (before) the unspecified groups (others).

Response

We corrected the order of the groups. The health care workers are now mentioned first.  -

In table 3, the percentages in "vaccine acceptability according to vaccine efficacy" does not refer to the total of the table but to the total of answering "yes" to acceptance of vaccine. Please make this clearer to the reader

Response

The percentages in "vaccine acceptability according to vaccine efficacy” were calculated with the total number of participants who would accept the vaccine as denominator.

Percentages of “reasons for vaccine hesitancy” were calculated with the total number of participants who would not accept the vaccine as denominator.

We now include this as a footnote of Table 3.

Reviewer 4 Report

The manuscript is well written and aims to investigate the state of acceptance of the anti-Covid19 vaccination in Mozambique.
The Authors analyse the partecipants characteristics and several variables influencing the population acceptability of the anti-Covid19 vaccine, between March 11th-20th 2021.
Since all the world is affected by the Sars-Cov2 pandemic, every data and information about it is highly needed in order to contain and fight the disease.
This work underline the importance of maximize the informations about vaccination efficacy and side effects in Mozambique population (but not only) in order to increase the population scientific knowledge and the acceptability to the new generation immunoprophylaxis.

I would like to appoint a few minor revision to the manuscript:
-Line 109: explain more in detail in the text the data and respective percentages reported in the Table1 below. "...majority were male (60%) and single (38.5%)". Please specify in the text if the "single" category is 
only referred  to males or to both males and females.

-Line113: report in the text the exact percentage written in the respective table. "..and in 29% (233).." 29% should be 28.8% instead.
Please do the same in line 125: 20% should be 20.2% instead, and in line130: 39% should be 38.6% instead.

-Table2: the sum of the participants's health characteristics is 1877 instead of 1879 (entire sample).

-
Lines 115-117: "Among the respondents ...positivity rate was similar among healthcare workers..and non-healthcare workers..". It should be specified in the text that the number of healthcare workers partecipants that took part of the online survey is less than the number of non-healthcare workers.

-Line 117: the sum of participants that reported to suffer from a chronic disease is Six hundred twenty-two instead of six hundred twenty-one.

-Lines 117-119: specify that hypertension and respiratory diseases are the most common chronic diseases affecting the partecipants, but not the only ones.

-Table3: specify on which pool of partecipants are calculated the percentages referred to "Reasons for vaccine hesitancy".

Author Response

Reviewer 4

The manuscript is well written and aims to investigate the state of acceptance of the anti-Covid19 vaccination in Mozambique.

The Authors analyse the participants characteristics and several variables influencing the population acceptability of the anti-Covid19 vaccine, between March 11th-20th 2021.

Since all the world is affected by the Sars-Cov2 pandemic, every data and information about it is highly needed in order to contain and fight the disease.

This work underline the importance of maximize the informations about vaccination efficacy and side effects in Mozambique population (but not only) in order to increase the population scientific knowledge and the acceptability to the new generation immunoprophylaxis.

I would like to appoint a few minor revision to the manuscript:

-Line 109: explain more in detail in the text the data and respective percentages reported in the Table1 below. "...majority were male (60%) and single (38.5%)". Please specify in the text if the "single" category is only referred to males or to both males and females.

Response

Single is referring to all study participants. We now state “ Most respondents were in the 18-35 age group (58.3%), majority were male (60.0%) and 38.5% were single (Table1).”

-Line113: report in the text the exact percentage written in the respective table. "..and in 29% (233).." 29% should be 28.8% instead.

Response

We corrected this

We changed these percentages

Please do the same in line 125: 20% should be 20.2% instead, and in line130: 39% should be 38.6% instead.

Response

We changed these percentages

-Table2: the sum of the participants' health characteristics is 1877 instead of 1879 (entire sample).

Response

Thanks for identifying this error. The total number of participants was 1878. We checked all numbers in the Tables and text of the paper and corrected the numbers if this was needed.

-Lines 115-117: "Among the respondents ...positivity rate was similar among healthcare workers and non-healthcare workers.". It should be specified in the text that the number of healthcare workers participants that took part of the online survey is less than the number of non-healthcare workers.

Response

We now state” Among respondents who reported COVID-19 test results, positivity rate was similar among the 324 healthcare workers (28.4%) and the 485 non-healthcare workers (29.1%); p=0.854.”

-Line 117: the sum of participants that reported to suffer from a chronic disease is Six hundred twenty-two instead of six hundred twenty-one.

Response

We corrected this

-Lines 117-119: specify that hypertension and respiratory diseases are the most common chronic diseases affecting the participants, but not the only ones.

Response

We now state “the most common were hypertension in 207 (11.0%) and a respiratory disease in 147 (7.8%) (Table 2).”

-Table3: specify on which pool of participants are calculated the percentages referred to "Reasons for vaccine hesitancy".

Response

Percentages were calculated with the total number of participants as denominator. We now mention this as a footnote in Table 3-
